# Endoscopic Treatment of Malignant Hilar Biliary Obstruction

**DOI:** 10.3390/cancers15245819

**Published:** 2023-12-13

**Authors:** Jakub Pietrzak, Adam Przybyłkowski

**Affiliations:** Department of Gastroenterology and Internal Medicine, Medical University of Warsaw, 02-091 Warsaw, Poland; jakub.pietrzak@uckwum.pl

**Keywords:** cholangiocarcinoma, malignant hilar biliary obstruction, endoscopic biliary drainage, endoscopic retrograde cholangiopancreatography

## Abstract

**Simple Summary:**

Biliary stenting is today the primary method of palliative and bridging treatment in patients with malignant hilar biliary obstructions. Systematization, collection and interpretation of the studies performed so far is necessary to form appropriate recommendations and guidelines for the management, selection of drainage methods, selection of appropriate types of stents, their quantity and possible additional methods of endoscopic treatment.

**Abstract:**

Stent implantation is an effective approach for palliative treatment of Bismuth-Corlette type III–IV malignant hilar biliary obstructions (MHBOs). In this article, we reviewed the currently used access methods for biliary stent placement (percutaneous transhepatic biliary drainage, endoscopic biliary drainage, endosonography guided biliary drainage), the available stent types (plastic stent, self-expanding metallic stent, full cover self-expanding metallic stent, radioactive self-expanding metallic stent), major approaches (unilateral, bilateral) and deployment methods (stent-in-stent, stent-by-stent). Finally, this review gives an outlook on perspectives of development in stenting and other palliative methods in MHBO.

## 1. Introduction

Malignant hilar biliary obstructions may be caused by cholangiocarcinoma, gallbladder carcinoma, hepatocellular carcinoma, pancreatic cancer or metastatic lymph node of liver hilum. The most common cause of malignant hilar biliary obstructions (MHBO) remains extrahepatic cholangiocarcinoma (CCA), which based on location, can be divided into perihilar CCA (accounts for ~50–70% of all CCA cases) and distal CCA. Bismuth-Corlette classification is used to classify the location of biliary strictures. The incidence in populations is variable; in Europe, it is between 0.5 and 3.4 per 100,000 people [1]. At the onset of disease symptoms, such as jaundice, pruritus, stool discoloration, dark urine, pain and cholangitis, most patients with Bismuth-Corlette III-IV stage are diagnosed at the inoperable stage. Median survival for inoperable cases ranges between 7 and 16 months [2]. Based on results of available studies, it is justified to conclude that biliary stenting not only relieves symptoms of obstruction but also prolongs survival time in MHBO [3,4,5].

## 2. Biliary Anatomy

Knowledge of biliary anatomy is essential for the technical success of the procedure and stent selection. The anatomy of the biliary tract is variable and sometimes complex, thus posing significant challenges for the optimal stent placement. In the most common anatomical variant [6,7], the right hepatic duct arises from the connection of the anterior and posterior sector bile ducts, which are responsible for the drainage of the 5th and 8th and 6th and 7th liver segments, respectively. Segments 2, 3 and 4 drain into the left bile duct, which in most cases is formed by connecting sector ducts 2 and 3 with the creation of the left lateral section duct, into which sector duct 4 enters. The longer length is generally characteristic of the left hepatic duct. There are various classifications which describe the anatomical variants of the biliary tract, the most commonly used are Nakamura [8], Varotti [9] and Huang [10] (Figure 1).

Averaging the volume of liver drainage through the bile ducts, the right hepatic duct is responsible for draining 50–60%, the left hepatic duct 30–40% and 10% of the liver volume from the caudate lobe, whose duct usually drains into the right hepatic duct.

The minimum liver drainage volume recommended by the European Society of Gastrointestinal Endoscopy (ESGE) and Asia-Pacific consensus [11] is ≥50%. Drainage of >50% of liver volume usually requires bilateral stenting. Vienne et al. [12] showed that drainage above 50% reduces the risk of cholangitis (OR 3.04, *p* = 0.01) and prolongs survival (119 vs. 59 days, *p* = 0.005).

There are no clear recommendations for disqualification of MHBO patients from biliary drainage, each case should be analyzed individually. It seems that the basic aspects indicating the lack of the expected effect of drainage may be the absence of obvious biliary dilatation, significant involvement of the liver parenchyma and poor general condition of the patient. 

## 3. Methods, Findings and Search Strategy

A search for randomized controlled trials [RCTs], case series, retrospective studies and meta-analyses has been performed. The PubMed(R) database (National Library of Medicine, Bethesda, MD, USA) was searched by keyword “malignant hilar biliary obstruction”, and the timeframe used in the search was from database inception until May, 2023. Included studies and case series covered patients diagnosed with malignant hilar biliary obstruction and treated with endoscopic or percutaneous biliary stenting. The initial search returned 671 articles, of which 48 studies met the inclusion criteria for this review. PRISMA plot for this review is available as Figure 2.

## 4. Access Methods

Both methods endoscopic biliary drainage (EBD) and percutaneous transhepatic biliary drainage (PTBD) have certain advantages as well as limitations. The choice of drainage method should depend on the localization of the obstruction, the clinical condition of the patient, the severity of the primary disease (ascites, gastrointestinal stenosis), and the level of experience in biliary drainage at each center.

American Society for Gastrointestinal Endoscopy (ASGE) 2021 [14] recommends endoscopic biliary drainage (EBD) as the first choice for potentially resectable MHBO, while for unresectable or palliative drainage, it recommends PTBD/EBD depending on patient preferences, disease characteristics and local expertise.

ESGE 2018 [15] recommends EBD if the preoperative drainage of MHBO is necessary (plastic stents or naso-biliary drains are preferred), while for unresectable or palliative drainage, ESGE recommends PTBD or a combination of PTBD and EBD.

Many studies to date have unequivocally found PTBD superior to EBD in terms of therapeutic success (Paik et al. [16], Moole et al. [17], Van Eecke et al. [18]), longer stent patency (Lee et al. [19]), incidence of overall complications, 30-day mortality, sepsis and duodenal perforation. In a recently published 10-year analysis, Páez-Carpio et al. [20] confirmed the safety and effectiveness of PTBD in MHBO with a technical success rate of 87.7%. Also, the largest meta-analysis comparing PTBD with EBD by Duan et al. [21] found a significantly lower incidence of cholangitis and pancreatitis in PTBD, with ORs of 0.48 (95% CI, 0.31 to 0.74) and 0.16 (95% CI, 0.05 to 0.52) for cholangitis and pancreatitis, respectively. Only for bleeding and stent dislocation the risk was higher for PTBD than EBD, with ORs of 1.81 (95% CI, 1.35 to 2.44) and 3.41 (95% CI, 1.10 to 10.60), respectively.

In clinical practice, EBD is a more frequently performed procedure than PTBD, despite slightly better success rates in favor of PTBD; however, endoscopic drainage seems to be definitely more patient-friendly, which is crucial in palliative drainage with the goal to improve quality of life. Moreover, EBD seems to be definitely the first choice method due to the higher risk of metastatic spread during PTBD (Wang et al. [22]). Despite the recent decline in the frequency of PTBD performed in favor of EBD, the success rate of technically difficult PTBD procedures is promising, forcing further development of this method. A recent example would be a case of the PTBD approach with fusion imaging of real-time ultrasonography and computed-tomography (Hosokawa et al. [23]) in a patient with MHBO (Bismuth IV), in whom right posterior sectional bile duct stenting was not feasible during EBD.

In summary, PTBD is mentioned as a first-line drainage method in guidelines, but in the majority of MHBO cases, ERCP is advised as the primary intervention. In the case of possible resectability, the presence of ascites, coagulation disorders, insufficient biliary dilatation and multiple liver metastases, ERCP should clearly be the first-choice method. However, in cases of an obstruction of the gastrointestinal tract or previous surgery that makes the biliary tract (Roux-en-Y) difficult to reach, PTBD should remain as the method of first choice. Failure of ERCP is also an indication for PTBD.

An alternative method of biliary access in MHBO is the endosonography-guided biliary drainage (EUS-BD). Sundaram et al. [24] and Winkler et al. [25] proved the usefulness of this method in MHBO, with satisfactory technical success and reasonable clinical success. The method involves the creation of a fistula with implantation of a stent between the stomach (EUS-guided hepaticogastrostomy) or duodenum (EUS-guided choldeochoduodenostomy). In the case of distal biliary obstruction, it is becoming the method of second choice, and numerous meta-analyses have confirmed its considerable efficacy (Jin et al. [26]). However, larger trials are still required for MHBO to include this method in the management pathway. It seems that it may become the primary method of access after ERCP failure, especially with the further development of tools and the continued growth of operators experience.

## 5. Stent Selection

There are two basic types of biliary stents: metal and plastic. Multiple sizes and shapes are available to accommodate physician preferences, different disease stage and patient anatomy. Among the plastic stents, there are straight stents, double pig tails with bent ends, perforated stents and self-removable stents. Self-expandable metal stents are divided into full covered self-expandable metal stents (FCSEMS) and uncovered self-expandable metal stents (UCSEMS). All stents are available in different diameters (3–12 mm) and lengths (40–120 mm).

Plastic stents, compared to metal stents, are characterized by smaller diameter and shorter patency time. A multicenter study by Xia et al. [27] revealed that the median symptom-free stent patency in the plastic stent placement group was 4.4 months, while in the SEMS group, the median symptom-free stent patency was 8.7 months. SEMS also achieved a better clinical success rate than plastic stents (90.8% vs. 68.6%, respectively, *p* < 0.001). The incidence of postoperative cholangitis was higher in the plastic stents group than in the SEMS group (27.9% vs. 13.0%, respectively, *p* < 0.001). An additional advantage of UCSEMS over the plastic stent is that UCSEMS does not block a side branch of the biliary tree (such as the cystic duct) (Table 1).

According to the classical approach, the average interval for scheduled plastic stent replacement should be no longer than 3 months. Despite the significantly lower cost of plastic stents, the overall cost-effectiveness may be adversely affected by frequent replacement.

ASGE [14] guidelines recommend that the choice of the stents should be based on the patient′s estimated survival time, desire to avoid reintervention and the possible lack of a definitive strategy. In the case of short life expectancy and a preference to avoid reintervention, UCSEMS are recommend, while when further management has not been fully established, the ASGE recommends implantation of a plastic stent with a possible replacement.

ESGE′s [15] recommendations are similar to those of the ASGE, in the absence of an established diagnosis, plastic stents are recommended, while for palliative drainage, the ESGE specifies the choice of UCSEMS. The ESGE’s guidelines mention the first retrospective study on FCSEMS efficacy in MHBO. Inoue et al. [33] showed a high technical success rate and a long time to recurrent biliary obstruction (210 days), however, liver abscesses were reported in 7% of patients as a complication of stent crossing a duct bifurcation. The guidelines do not specify or recommend the use of FCSEMS.

For preoperative drainage, the ASGE and ESGE guidelines appear to be adequate. A recently published study comparing FCSEMS with plastic stent (Mori et al. [34]) showed no difference for RBO until surgery, while it confirmed fewer postoperative complications with plastic stents. Regarding the method of stent placement for preoperative drainage, above the papilla or inside the bile duct, Ishiwatari et al. [35] in a recent retrospective multicenter study showed comparable results in these two methods. In contrast, She et al. [36] and Hameed at al. [37] compared the effect of the choice of preoperative drainage method (ERCP, PTBD, combination of ERCP and PTBD) on postoperative outcomes, finding no significant differences.

It seems that in non-operative cases, the role of FCSEMS in MHBO is underestimated. With recent developments in chemotherapy and prolonging survival of patients with MHBO (for example with CCA) stenting with UCSEMS is becoming a suboptimal strategy due to tumor overgrowth by the wire mesh of UCSEMS and the inability to replace the stent (Table 2.). 

To date, there have been only five studies published evaluating the use of FCSEMS in MHBO (Inoue et al. [33], Yoshida et al. [38], Kitamura et al. [39], Takahashi et al. [40], Matsubara et al. [41]). Each one has shown high technical and clinical success rates and the high success rate of endoscopic re-intervention. Inoue et al. [26] report the highest time to stent patency of all the papers (210 days in the initial group after bilateral placement and 112 days and 152 days in the re-intervention group after bilateral and unilateral placements, respectively). Intrahepatic bile duct occlusions and stent migration appear to be the greatest limitation of the method. Virtually all available studies were performed with too small number of patients to definitively determine the efficacy of the method (17, 32, 17, 54, 11 patients, respectively). Large prospective studies are needed to evaluate the method.

## 6. Stent Placement Strategy (Unilateral, Bilateral, Trisegmental) 

The multicenter study comparing unilateral vs. bilateral stenting in MHBO by Xia et al. [27] showed better jaundice control, longer stent patency (8.1 months [95% CI, 6.8–9.4] vs. 5.4 months [95% CI, 4.7–6.2]; P Z 0.018) and longer overall survival (5.2 months [95% CI, 4.6–5.8] vs. 4.0 months [95% CI, 3.3–4.7]; P Z 0.040) in favor of bilateral stenting. Also, one of the larger meta-analyses by Chen et al. [42] confirms better clinical success rates (odds ratio: 3.56; 95% CI: 1.62–7.82, *p* = 0.002) and a reduced incidence of stent dysfunction (odds ratio: 0.51; 95% CI: 0.30–1.00, *p* = 0.05) in patients undergoing bilateral stenting (Table 3). Despite more favorable results for bilateral stenting, the issue is still unresolved. The studies conducted to date (e.g., Xia et al. [27]), despite the large number of patients and the use of propensity score matching (PSM), are retrospective papers evaluating different diseases at different stages of development.

It is technically more difficult to implant two stents than to perform unilateral stenting (Yang et al. [43]). Bilateral stenting prevents accidental blockage of bile outflow from the non-stented hepatic ducts. If technically possible, bilateral metal stent placement is preferred.

**Table 3 cancers-15-05819-t003:** Studies on unilateral and bilateral stenting in malignant hilar biliary obstruction.

	Type, Number of Patients	Clinical Success Rate (%)	Median Time to RBO (Days)	Rates or Mean Number of Reintervention	The Incidence of Post-ERCP Cholangitis
Naitoh et al. (2009) [44]	bi, 29; uni, 17	96%, 100%	488 210	- -	3% 0%
Lee et al. (2017) [45]	bi, 67; uni, 66	95.3%, 84.9%	252 193	42.2% 57.6%	10.4% 10.6%
Teng et al. (2019) [46]	bi, 52; uni, 58	98%, 96%	198 182	- -	- -
Staub et al. (2020) [47]	bi, 137; uni, 50	82.5%, 86%	168 158	- -	6.3% 0%
Xia et al. (2020) [27]	bi, 178; uni, 178	84.8%, 75.3%	246.5 164.4	1.7 +/− 1.2 1.8 +/− 1.1	17.4% 23%

Stent-by-stent (SBS) involves the simultaneous placement of two stents next to each other, depending on technical feasibility either into sectoral conduits or intrahepatic conduits. This method allows for multiple choices of stents, for example, two UNSEMS, two FCSEMS, two plastic stents or a combination between plastic and FCSEMS/UCSEMS.

Stent-in-stent (SIS) involves the placement of two UCSEMS with one stent crossing the other more than halfway through the stent and passing through a wire mesh. Both methods have their pros and cons; in general, SIS seems to be a method resembling natural drainage, but it is associated with the impossibility of eventual replacement or removal of the stents. Available studies and meta-analyses show different results with regard to the clinical success, complications, stent dysfunction and technical success.

Meta-analysis by Cao et al. [48] showed a marginally better success rate in the SIS method compared to SBS, with no significant differences for clinical success, complications and stent dysfunction. Also, Hong et al. [49], Kim et al. [50] and Ishigaki et al. [51] found similar results in both methods. In contrast, de Souza et al. [52] in their meta-analysis found longer stent patency for SIS compared to SBS and no differences in technical success, clinical success, rates of both early and late adverse events, reintervention and procedure-related mortality.

In cases of advanced Bismuth III-IV hilar obstruction, obtaining satisfactory drainage may require trisegmental drainage. A multicenter retrospective study comparing bilateral and trisegment drainage has recently emerged. Matsumoto et al. [53] found no statistically significant difference for stent patency but observed a significant difference in clinical success rates for reinterventions with trisegmental drainage (73% [11/15] vs. 96% [47/49], *p* = 0.009). It seems that with the development of the slim delivery system and novel endoscopic technique (Maruki et al. [54]), the placement of trisegmental drainage will increase.

## 7. Additional Palliative Therapies

Radiation-emitting metallic stents (REMS) are a combination of uncovered metal stents with brachytherapy realized by multiple I^125^ seeds. The main assumption of using REMS is not only to decongest the biliary tract but also a reduction in the tumor mass. To date, there have been several studies or case series published evaluating the efficacy and safety of REMS placement in MHBO. The results so far are very encouraging. REMS seem to prolong survival as well as effective biliary drainage in these patients (Lu et al. [55]). Also, one of the largest meta-analyses performed by Huang et al. [56] comparing the regular and radiation-emitting SEMS showed that REMS insertion is associated with longer overall survival and stent patency in patients with inoperable MHBO. Despite the proven efficacy of REMS, their position in MHBO treatment and widespread use remains questionable, thus additional multicenter clinical trials are expected in order to determine the appropriate indications. Additionally, the still limited access to nuclear medicine facilities remains a limitation of the method. (Table 4).

Radiofrequency ablation (RFA) is a method involving cancer cell reduction through high temperature achieved by high-frequency radio-waves. High temperature (>60 °C) generated by radio-waves causes denaturation of proteins which leads to coagulation and necrosis. In general, RFA is a method with proven efficacy in the palliative treatment of MHBO. To date, there have been several studies published evaluating its effectiveness. Of special note is a meta-analysis performed by Sofi et al. [60], comparing RFA with metallic or plastic stent placement (*n* = 239) or biliary stent placement only (*n* = 266). This meta-analysis showed prolonged survival (285 vs. 248 days) and improved stent patency in the group of patients treated with RFA. However, RFA was associated with a higher rate of adverse events, such as abdominal pain (31% vs. 20%, *p* = 0.003) (Table 5.).

An interesting aspect is the use of RFA in patients with originally implanted SEMS who have had tumor ingrowth through the stent wire mesh. Kadayifci et al. [65] compared 25 patients with an occluded SEMS treated with RFA and 25 patients after plastic stent placement only. The study found a significantly longer time of stent patency in the RFA group compared to the plastic stent placement only (119.5 vs. 65.3 days, *p* = 0.03).

In conclusion, RFA is a method with proven efficacy, but it seems inadequate as a single method of biliary decongestion in MHBO, while it may be recommended in patients with recurrent stenosis after primary SEMS implantation.

## 8. Conclusions

The multitude of available methods confirms the recent development (the introduction of the new delivery system and RFA) in the field of endoscopic treatment of MHBO. Due to the increasing number of patients with MHBO, further development of endoscopic techniques is required. Future perspectives in endoscopic biliary treatment involve several areas that aim to prevent stent occlusion, facilitate stent implantation, enhance ablation techniques and ultimately improve patient outcomes. 

## Figures and Tables

**Figure 1 cancers-15-05819-f001:**
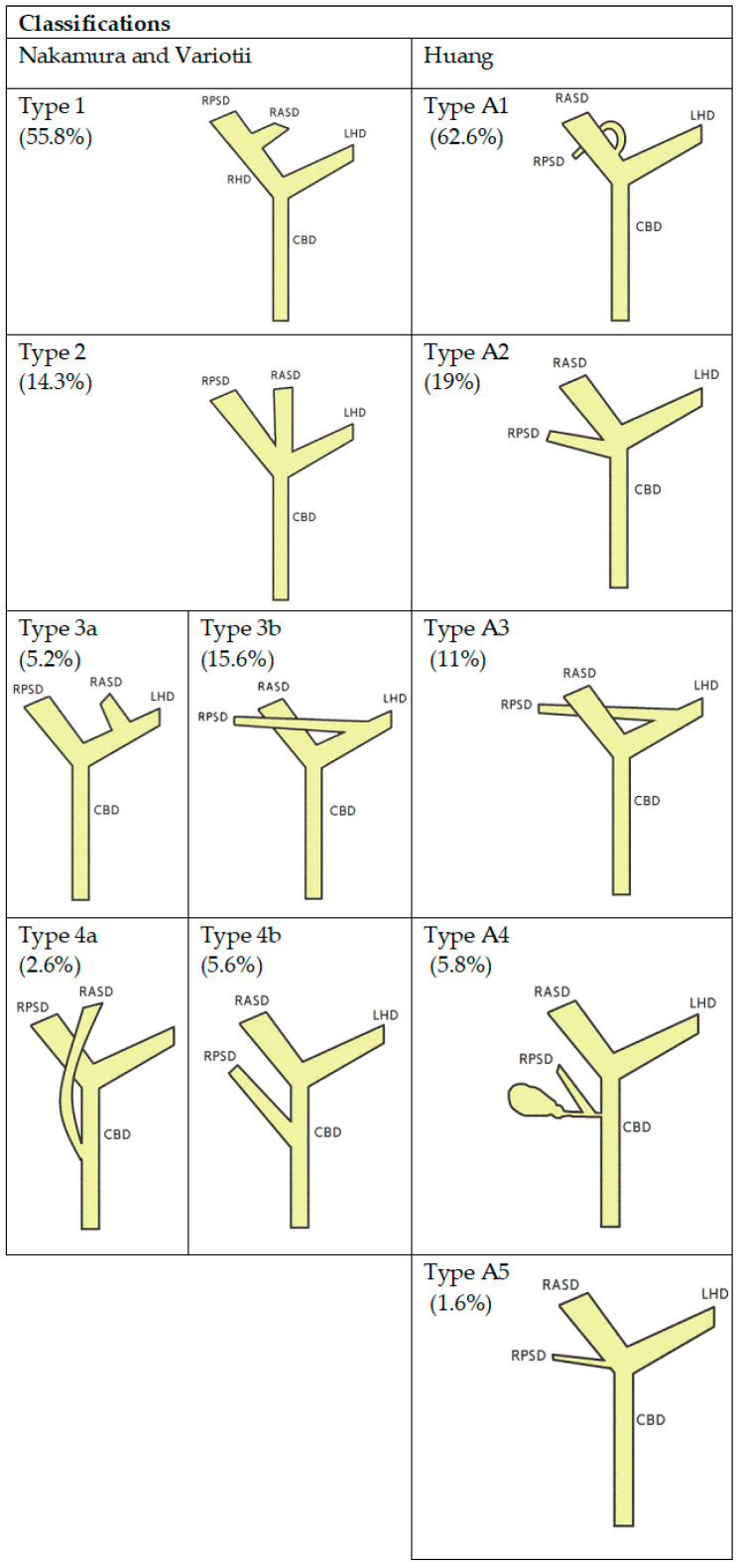
Schematic diagrams of the anatomical variants of the bile ducts according to Nakumra, Variotti, Huang and respective frequencies (in parenthesis). RPSD—right posterior sectoral duct, RASD—right anterior secoral duct, RHD-right hepatic duct, LHD—left hepatic duct, CBD—common biliary duct.

**Figure 2 cancers-15-05819-f002:**
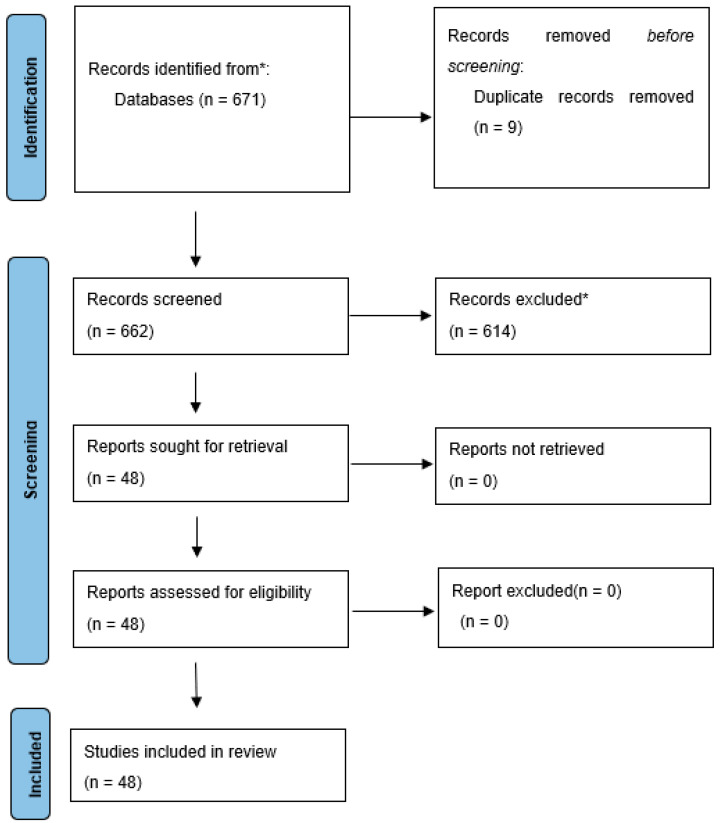
Malignant hilar biliary obstruction (PRISMA)**.** From: Page et al. [13]. * Reasons for excluding records or full-text articles were as follows: publications non concerning endoscopic or percutaneous methods of biliary drainage or studies on distal or benign biliary obstruction, absence of abstract, review papers topic, articles in a language other than English, single case reports.

**Table 1 cancers-15-05819-t001:** Studies on uncovered metal stents and plastic stents in malignant hilar biliary obstruction.

	Type, Number of Patients	Clinical Success Rate (%)	Median Time to RBO (Day)	Mean Number of Reintervention	The Incidence of Post-ERCP Cholangitis
Wagner et al. (1993) [28]	USEMS, 11; plastic, 9	100% 88.9%	- -	2.4 +/− 2.6 1.1 +/− 0.8	9.1% 33.3%
Liberato et al. (2012) [29]	USEMS, 246; plastic, 204	97.9% 84.8%	189 140	- -	5.7% 33.3%
Sangchan et al. (2013) [30]	USEMS, 54; plastic, 54	70.4% 46.3%	103 35	1.16 1.23	14.8% 24%
Gao et al. (2017) [31]	USEMS, 28; plastic, 31	92.9% 93.5%	119 93	- -	10.7% 12.9%
Xia et al. (2020) [27]	USEMS, 184; plastic, 172	90.8% 68.6%	264.8 133.9	1.5 +/− 0.7 2.2 +/− 1,4	13.0% 27.9%
Kim et al. (2021) [32]	USEMS, 35; plastic, 64	71.4% 65.6%	112 56	- -	significantly higher in the plastic group

USEMS—uncovered metal stent; RBO—recurrent biliary obstruction.

**Table 2 cancers-15-05819-t002:** Studies on full cover metal stents in malignant hilar biliary obstruction.

	Method, Number of Patients	Technical Success Rate (%)	Clinical Success Rate (%)	Percentage of RBO (%)	Median Time to RBO (Day)	Success Rate of Reintervention (%)	The Incidence of Post-ERCP Cholangitis
Inoue et al. (2016) [33]	SBS, 17	94%	100%	31%	210	100%	5.8%
Yoshida et al. (2016) [38]	SBS, 32	96.9%	93.5%	61%	95	83.3%	0%
Kitamura et al. (2017) [39]	SBS, 17	100%	82%	71%	79	46%	5.8%
Takahashi et al. (2022) [40]	SBS, 54	100%	92.5%	35.2%	181	100%	1.8%
Matsubara et al. (2022) [41]	SBS, 11	100%	100%	36.4%	187	100%	0%

SBS—stent-by-stent; RBO—recurrent biliary obstruction.

**Table 4 cancers-15-05819-t004:** Studies on radiation-emitting metallic stent in malignant hilar biliary obstruction.

	Type, Number of Patients	The Median Overall Survival (Days)	The Median Stent Patency (Days)	The Clinical Success Rate (%)	The Incidence of Overall Complications (%)
Lu et al. (2017) [55]	REMS, 33 UCSEMS, 26	338, 141	385 142	87.9% 84.6%	27.3% 26.9%
Zhou et al. (2020) [57]	REMS, 40 UCSEMS, 36	177, 123	387 121	95% 97.2%	50% 38.9%
Chen et al. (2021) [58]	REMS, 36 UCSEMS, 48	250, 188	225 165	100% 93.3%	19.4% 22.9%
Zhang et al. (2023) [59]	REMS, 34 UCSEMS, 30	405, 264	- -	94.1% 93.3%	11.8% 10%

REMS—radiation-emitting metallic stent; UCSEMS—uncovered metal stent.

**Table 5 cancers-15-05819-t005:** Studies on radiofrequency ablation in malignant hilar biliary obstruction.

	Type, Number of Patients	The Median Overall Survival (Days)	The Median Stent Patency (Days)	The Clinical Success Rate (%)	The Incidence of Post-ERCP Cholangitis (%)
Sofi et al. (2017) [60] (meta-analysis)	RFA, 239 stent, 266	285 248	Mean time 50.6 longer in the RFA group	100% 93.3%	- -
Han et al. (2020) [61]	RFA, 16	147	90	100%	6.3%
Inoue et al. (2020) [62]	RFA, 41	244	230	95.1%	2.5%
Kang et al. (2022) [63]	RFA, 15 stent, 15	230 144	178 122	100% 93.3%	- -
Oh et al. (2022) [64]	RFA, 28 stent, 51	311 311	140 192	100% 100%	- -

RFA—Radiofrequency ablation.

## Data Availability

No new data were created or analyzed in this study. Data sharing is not applicable to this article.

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
