# Peer review of "Endoscopic Treatment of Malignant Hilar Biliary Obstruction"

_cancers, 2023, doi:10.3390/cancers15245819_

Round 1
Reviewer 1 Report
Comments and Suggestions for Authors
The authors has reviewed and described the biliary drainage approach, stent type, novel stent, bilateral and trisegmental drainage, stent deployment method, and adjunctive methods such as RFA in MHBO. A few typos need to be corrected.
> Minor points>
>
> 1. Page 1 Line 8
>
> Implanatation -> implantation
>
>
> 2. Page 1 Line 22
>
> MHBO -> malignant hilar biliary obstruction (MHBO}
>
>
> 3. Page 1 Line 23
>
> “HCCA” should be described in full words at first. And “hilar” seems
> to be redundant as well.
>
>
> 4. Page 1 Line 28
>
> Median survival for inoperable perihilar cholangiocarcinoma seems to
> be too short. Please add the reference.
>
>
> 5. Page 4 Line 112
>
> ACCES -> ACCESS
Author Response
Dear Reviewer!
Thank you very much for taking the time to review this manuscript. Please find the detailed responses below and the corresponding revisions in the re-submitted files.
- Point-by-point response to Comments and Suggestions for Authors
> 1. Page 1 Line 8
>
> Implanatation -> implantation
. We have corrected a typo.
>
> 2. Page 1 Line 22
>
> MHBO -> malignant hilar biliary obstruction (MHBO}
All abbreviations that first appear have been spelled-out.
>
> 3. Page 1 Line 23
>
> “HCCA” should be described in full words at first. And “hilar” seems
> to be redundant as well.
All abbreviations have been spelled-out. “hilar” has been deleted.
>
> 4. Page 1 Line 28
>
> Median survival for inoperable perihilar cholangiocarcinoma seems to
> be too short. Please add the reference.
We changed the median value and added references
>
> 5. Page 4 Line 112
>
> ACCES -> ACCESS
We have corrected a typo.
I hope that the amendments will make the manuscript meet the criteria for acceptance for
publication.
Jakub Pietrzak
Reviewer 2 Report
Comments and Suggestions for Authors
Manuscript ID: cancers-2727495
Title: Endoscopic treatment of malignant hilar biliary obstruction
Manuscript summary: The authors reviewed various palliative treatment methods in malignant hilar biliary obstruction and summarized recent trend in stent placement.
The reviewer has no conflict of interest.
General impression:
Strong point: This manuscript covers recent update of biliary stent
.
Specific comments
1. Title: Okay
2. Abstract: P1 L8 stent implanatation -> stent implantation
3. Page1 Line22, 23: MHBO, HCCA; abbreviations should be spelled-out when those first appear throughout this manuscript. Abstract, body, table, figure, respectively.
4. P1 L32 It would be better to make a new heading as biliary anatomy. The Review article does not need to follow materials and methods, results section. The guidelines suggest that review include Introduction, Relevant Sections, Discussion, Conclusions, and Future Directions. It is recommended to re-organize headings.
5. P1 L39 “The longer length is generally characteristic for the left hepatic duct which make it a target for decompression in cases of malignant obstruction”. This sentence requires revision because the priority for decompression side should be decided on liver volume and portal vein patency, not on duct length.
6. Table 2 legend: not (in brackets), but in parenthesis
7. P4 L112 Please correct typo of ACCES METHOD as ACCESS METHOD.
8. P5 L118 spell-out ASGE, ESGE
9. P5 L133 There is redundant phase of ‘for bleeding and stent dislocation’
10. P6 L178 UCSESM -> UCSEMS
11. P7 L225 Please correct typo of ‘MAIOR’ as MAJOR.
12. MAJOR APPROCHES seems not well represent the contents, the reviewer suggests STENT PLACEMENT STRATEGY or STENTING STRATEGY.
13. P7 L225 Regarding unilateral vs. bilateral, it is worthy to note that unilateral vs bilateral stenting is still controversial. Even Xia et al. study include large numbers of patients and PSM, the study is retrospective and has limitations. In unilateral stenting, there can be a strategic one-side drainage and a technical problem. Moreover, the study included many heterogeneous disease and various stages.
14. P7 L235 dual -> bilateral
15. Table 5 Mean number of reintervention -> Rates or mean number of reintervention
16. Table 5 Add legends for bi, uni, and RBO
17. P8 L260 the place -> the placement
18. P8 L262 Radioactive stents (REMS) -> radiation emitting metallic stent (REMS)
19. P9 L278 C -> Celsius, or ℃
20. Table 6 Spell-out REMMS and UCSEMS in table legend.
21. There is mixed use of Arabic and roman numerals in citations, especially in tables. Please change reference number as Arabic numerals, not roman.
22. Check the citation and reference number. There are several discrepancies, eg. Inoue et al. 2016 (26) in Table 4, Xia et al (2020) (25) in table 3.
23. P8 L255 Typo: tridegmental -> trisegmental
Summary
This manuscript describes well about the recent trend of palliative treatment of malignant hilar biliary obstruction.
Comments on the Quality of English LanguageMinor spell and grammar checks were required. Overall, it is well-written.
Author Response
Dear Reviewer!
Thank you very much for taking the time to review this manuscript. Please find the detailed responses below and the corresponding revisions in the re-submitted files.
- Point-by-point response to Comments and Suggestions for Authors
- Title: Okay
Thank you.
- Abstract: P1 L8 stent implanatation -> stent implantation
We have corrected a typo.
- Page1 Line22, 23: MHBO, HCCA; abbreviations should be spelled-out when those first appear throughout this manuscript. Abstract, body, table, figure, respectively.
We have revised. All abbreviations that first appear have been spelled-out.
- P1 L32 It would be better to make a new heading as biliary anatomy. The Review article does not need to follow materials and methods, results section. The guidelines suggest that review include Introduction, Relevant Sections, Discussion, Conclusions, and Future Directions. It is recommended to re-organize headings.
We have partially changed the headings. We left the section "METHODS, FINDINGS AND SEARCH STRATEGY" because of the authentication of the systematic review according to the PRISMA protocol
- P1 L39 “The longer length is generally characteristic for the left hepatic duct which make it a target for decompression in cases of malignant obstruction”. This sentence requires revision because the priority for decompression side should be decided on liver volume and portal vein patency, not on duct length
We have revised. Modified – “The longer length is generally characteristic for the left hepatic duct”
- Table 2 legend: not (in brackets), but in parenthesis
Language error removed
- P4 L112 Please correct typo of ACCES METHOD as ACCESS METHOD.
We have corrected a typo.
- P5 L118 spell-out ASGE, ESGE
We have revised. All abbreviations that first appear have been spelled-out.
- P5 L133 There is redundant phase of ‘for bleeding and stent dislocation’
The phrase has been removed
- P6 L178 UCSESM -> UCSEMS
We have corrected a typo.
- P7 L225 Please correct typo of ‘MAIOR’ as MAJOR.
We have corrected a typo.
- MAJOR APPROCHES seems not well represent the contents, the reviewer suggests STENT PLACEMENT STRATEGY or STENTING STRATEGY.
We changed on „STENT PLACEMENT STRATEGY (unilateral, bilateral, trisegmental)”
- P7 L225 Regarding unilateral vs. bilateral, it is worthy to note that unilateral vs bilateral stenting is still controversial. Even Xia et al. study include large numbers of patients and PSM, the study is retrospective and has limitations. In unilateral stenting, there can be a strategic one-side drainage and a technical problem. Moreover, the study included many heterogeneous disease and various stages.
We added the statement – “Despite more favorable results for bilateral stenting, the issue is still unresolved. The studies conducted to date (e.g., Xia et al.), despite the large number of patients and the use of propensity score matching (PSM), are retrospective papers evaluating different diseases at different stages of development.”
- P7 L235 dual -> bilateral
We have corrected a typo.
- Table 5 Mean number of reintervention -> Rates or mean number of reintervention
We corrected the text in the table.
- Table 5 Add legends for bi, uni, and RBO
We have expanded the abbreviations in the table legend.
- P8 L260 the place -> the placement
We have corrected a typo.
- P8 L262 Radioactive stents (REMS) -> radiation emitting metallic stent (REMS)
We have corrected a typo.
- P9 L278 C -> Celsius, or ℃
We changed on ℃
- Table 6 Spell-out REMMS and UCSEMS in table legend.
We have expanded the abbreviations in the table legend.
- There is mixed use of Arabic and roman numerals in citations, especially in tables. Please change reference number as Arabic numerals, not roman.
We changed all numbers in tables to Arabic
- Check the citation and reference number. There are several discrepancies, eg. Inoue et al. 2016 (26) in Table 4, Xia et al (2020) (25) in table 3.
All citations have been checked and corrected
- P8 L255 Typo: tridegmental -> trisegmental
We have corrected a typo.
I hope that the amendments will make the manuscript meet the criteria for acceptance for
publication.
Jakub Pietrzak
Reviewer 3 Report
Comments and Suggestions for Authors
Pietrzak and a co-worker presented a review on the Endoscopic treatment of malignant hilar biliary obstruction. The authors have focussed on methods for biliary stent placement, stent types, and an overview of development in stenting and other palliative methods. This review opens the avenue to alternate approaches in the endoscopic treatment of malignant HBO.
1. Plagiarised texts to be modified for 2nd paragraph: first half part has plagiarised text.
2. Table 1 is actually a flow chart/figure only. It's not a real table. so ,modify, accordingly.
3. Tables 1, 2, 3.. must be organized sequentially and must b mentioned in an appropriate position in the text.
4. Authors must have distinguish tables and figures and charts, clearly.
5. Conclusions must be elaborated and must include future perspectives.
6. All the references are to be formatted and cited as per the journal-specific guidelines.
Author Response
Dear Reviewer!
Thank you very much for taking the time to review this manuscript. Please find the detailed responses below and the corresponding revisions in the re-submitted files.
- Point-by-point response to Comments and Suggestions for Authors
- Plagiarised texts to be modified for 2nd paragraph: first half part has plagiarised text.
We've conducted a thorough check using an anti-plagiarism test and found no plagiarised text. I've attached a report from the anti-plagiarism survey for your reference. However as the anti-plagiarism test could be not sensitive enough please indicate the exact location of the alleged plagiarized content and we will rephrase it.
- Table 1 is actually a flow chart/figure only. It's not a real table. so ,modify, accordingly.
We have edited the names of figures and tables.
- Tables 1, 2, 3.. must be organized sequentially and must b mentioned in an appropriate position in the text.
Figures and tables have been placed in appropriate places in the text.
- Authors must have distinguish tables and figures and charts, clearly.
We have edited the names of figures and tables.
- Conclusions must be elaborated and must include future perspectives.
Conclusions have been corrected.
- All the references are to be formatted and cited as per the journal-specific guidelines.
Citations have been reformatted according to the journal's guidelines
I hope that the amendments will make the manuscript meet the criteria for acceptance for publication.
Jakub Pietrzak

Round 2
Reviewer 1 Report
Comments and Suggestions for Authors
Many thanks for the aurthors' efforts to revise the manuscript.
Author Response
Dear Reviewer!
Thank very much for your help and kindness
Yours faithfuly
Jakub Pietrzak
Reviewer 2 Report
Comments and Suggestions for Authors
Manuscript ID: cancers-2727495
Title: Endoscopic treatment of malignant hilar biliary obstruction
Manuscript summary: The manuscript was well amended to the reviewer’s comments.
Specific comments/suggestions
1. P1L9 MHBO -> malignant hilar biliary obstructions
2. P1L31 “At the onset of disease symptoms, such as jaundice, pruritus, stool discoloration, dark urine, pain, and cholangitis, most cases are diagnosed at the inoperable stage, Bismuth-Corlette III-IV stage”: suggestion -> At the onset of disease symptoms, such as jaundice, pruritus, stool discoloration, dark urine, pain, and cholangitis, most patients with Bismuth-Corlette III-IV stage are diagnosed at the inoperable stage.
3. P3L113 Figure 2 There is a missing part in flow chart.
4. P4L202 Please use correct full term: SEMS -> self-expandable metal stents
5. P7L303 Radiation emitting metallic stent -> Radiation-emitting metallic stent: please change it in all the followings.
6. P8L319 please remove ‘degrees’
Summary
The reviewer recommends minor changes.
Author Response
Dear Reviewer!
Thank very much for your help and kindness. All suggestion are obviously endorsed and the errors have been corrected as detailed below:
> 1. P1L9 MHBO -> malignant hilar biliary obstructions
>
> MHBO -> malignant hilar biliary obstruction (MHBO)
All abbreviations that first appear have been spelled-out.
> 2. P1L31
>
> “At the onset of disease symptoms, such as jaundice, pruritus, stool discoloration, dark urine, pain, and cholangitis, most cases are diagnosed at the inoperable stage, Bismuth-Corlette III-IV stage” -> “At the onset of disease symptoms, such as jaundice, pruritus, stool discoloration, dark urine, pain, and cholangitis, most patients with Bismuth-Corlette III-IV stage are diagnosed at the inoperable stage.”
We have changed the wording of the sentence as proposed
> 3. P3L113 Figure 2 There is a missing part in flow chart.
>
We have completed the missing part of Figure 2.
> 4. P4L202 Please use correct full term: SEMS -> self-expandable metal stents
>
The abbreviation has been expanded
>
> 5. P7L303 Radiation emitting metallic stent -> Radiation-emitting metallic stent: please change it in all the followings.
>
We have changed throughout the article
> 6. P8L319 please remove ‘degrees’
>
We have removed the word “degrees”.
Once more I would like to thank you for your time and patience.
Yours faithfully
Jakub Pietrzak
Reviewer 3 Report
Comments and Suggestions for Authors
The authors have made necessary corrections and the manuscript can be accepted for publication.
Author Response

(The authors gave the same response as above.)
